# Stellar Turbulent Convection: The Multiscale Nature of the Solar Magnetic Signature

**Stefano Scardigli** , **Francesco Berrilli \*** , **Dario Del Moro** and **Luca Giovannelli**

Physics Department, University of Rome Tor Vergata, I-00133 Rome, Italy; s.scardigli@gmail.com (S.S.); dario.delmoro@roma2.infn.it (D.D.M.); luca.giovannelli@roma2.infn.it (L.G.)
\* Correspondence: francesco.berrilli@roma2.infn.it

**Abstract:** The multiscale dynamics associated with turbulent convection present in physical systems governed by very high Rayleigh numbers still remains a vividly disputed topic in the community of astrophysicists, and in general, among physicists dealing with heat transport by convection. The Sun is a very close star for which detailed observations and estimations of physical properties on the surface, connected to the processes of the underlying convection zone, are possible. This makes the Sun a unique natural laboratory in which to investigate turbulent convection in the hard turbulence regime, a regime typical of systems characterized by high values of the Rayleigh number. In particular, it is possible to study the geometry of convection using the photospheric magnetic voids (or simply voids), the quasi-polygonal quiet regions nearly devoid of magnetic elements, which cover the whole solar surface and which form the solar magnetic network. This work presents the most extensive statistics, both in the spatial scales studied (1–80 Mm) and in the temporal duration (SC 23 and SC 24), to investigate the multiscale nature of solar magnetic patterns associated with the turbulent convection of our star. We show that the size distribution of the voids, in the 1–80 Mm range, for the 317,870 voids found in the 692 analyzed magnetograms, is basically described by an exponential function.

**Keywords:** multiscale convection; stellar convection; turbulent convection; solar convection; solar magnetic field; solar granulation; solar mesogranulation; solar supergranulation; image processing

## 1. Introduction

The energy flux through the stellar interior is one of the fundamental quantities that govern the stellar structure. This flux is largely convective in the outermost layers of Sun-like stars (e.g., [1]) and becomes convective throughout the whole stellar structure in low-mass stars and brown dwarfs (e.g., [2]). Therefore, the heat transport and dynamics in these **convection** zones (CZs) are defined by organized flows of plasma. These flows are believed to contribute substantially to developing and maintaining the global stellar magnetic dynamo process, the process that is responsible for the cyclic regeneration of the stellar/solar large-scale magnetic field, which is at the basis of all those processes known as stellar activity (e.g., [3,4]).

While researchers have reached consensus on the basic ingredients needed to trigger the global stellar dynamo process (see again, [4]), how exactly this dynamo works remains unclear (e.g., [5–8]). Furthermore, it should be remembered that in stellar CZs the Rayleigh number ($Ra$) typically exceeds the value of $10^{15}$ (e.g., [9–11]). We recall that $Ra$ quantifies the relative magnitude of thermal forcing to dissipative forces in the fluid convective flows. Indeed, it represents the buoyancy forcing divided by the product of the rate of heat diffusion and viscous drag, at least in the rather crude Boussinesq convection (i.e., nearly incompressible plasma) approximation in the outermost stellar CZs. The expression for $Ra$ is given by:

$$Ra = \frac{g\alpha\Delta T L^3}{\nu\kappa} \qquad (1)$$

with $g$ the local gravity, $\alpha$ the thermal expansion coefficient, $\Delta T$ the temperature difference between top and bottom CZs, $L$ the CZ thickness, $\nu$ the kinematic viscosity and $\kappa$ the plasma thermal diffusivity. $Ra$ is estimated to be of the order of $10^{23}$ in the outer part of the Sun (e.g., [12]). The high $Ra$ value expected in stellar CZs has two main effects. The first is that the convection becomes strongly turbulent [13]. This regime has important implications on the structuring of convective cells and on the dynamic properties of temperature distribution within the fluid (for details refer to [14,15]). The second, of paramount importance in stars, but more broadly in all physical systems with high $Ra$, involves the efficiency in the transport of energy by convection. This efficiency can be quantified by the Nusselt number, $Nu$, which is the non-dimensional ratio of the total energy flux to the conductive (radiative) flux allowed for the same $\Delta T$ between top and bottom CZs.

When looking for a functional relationship between $Nu$ and $Ra$, simple scaling arguments supported by laboratory experiments have suggested that $Nu \propto Ra^{\beta}$. However, although for $\beta$ there are both theoretical evaluations, based on stability arguments and dimensional considerations, and estimates from laboratory experiments under various experimental conditions, the basic properties of heat transport in the hard turbulence regime remain unclear. In particular, it is not clear whether there exists an asymptotic regime that is supposed to occur at very high $Ra$. Indeed, as $Ra$ increases, displacements are observed from a basic assumption that $Nu$ scales as $Ra^{\frac{1}{3}}$ and this indicates the need for further analyses and experiments to clarify the heat transfer efficiency of the turbulent convective flows in high $Ra$ regimes (e.g., [16]).

In addition to these basic uncertainties in the physics governing turbulent convection, we must be aware that in the stellar case there are further possible complications. In fact, in this case, we must take into account that the dynamics of the plasma, stratified and compressible in the outermost regions of the star, is typically modified by the fact that the plasma flows interact with or even generate magnetic fields and, possibly, by the influence of the Coriolis force due to the stellar rotation.

This theoretical and experimental difficulty of quantifying the dependence of $Nu$ on $Ra$ in physical systems characterized by stratified plasma in the magneto-convective regime and very high $Ra$ makes the Sun the ideal laboratory for these studies.

Indeed, the Sun is the only star for which detailed observations and estimations of physical properties on the surface and in the CZ are currently possible and represents a unique natural laboratory in which to study turbulent convection in the hard turbulence regime. This is especially the case since spatial, temporal and spectral high-resolution observations of its photosphere are possible from ground-based (e.g., [17–22]) or space (e.g., [23,24]) instruments.

Detailed observation of the Sun's surface and its scaling properties related to temperature, velocity or magnetic structures raises several interesting questions about pattern formation in turbulent convection conditions at high $Ra$ (e.g., [25–32]). For many of these questions there are still no certain answers. For example, what determines the distribution of size and average life (e.g., [21,31,33]) of convective structures? Are the classic scales such as granulation, mesogranulation and supergranulation real convective scales or do they instead emerge from self-organization of surface flows (e.g., [27,34]) or from methods of observation of the sun (e.g., [28,35]), i.e., emergent structures/patterns resulting from physical processes sampled, and consequently filtered, at particular spatial and temporal scales? How is the persistence of structures organized for many tens of hours (e.g., [26,31,36]) possible in a regime of hard turbulence convection?

Photospheric magnetic voids (PMVs) offer a unique opportunity for studying convective signatures in a physical system governed by very high $Ra$ [37,38].

PMVs, or simply voids, are multiscale (Mm to tens of Mm), quasi-polygonal quiet-sun regions nearly devoid of magnetic elements, which cover the whole solar surface [39,40] and which form the well-known magnetic network typically associated with the chromo-

spheric network (e.g., [41–46]). The relative absence of a photospheric magnetic field in these regions is due to the advective transport process generated by convective flows.

Discussing the origin of the photospheric magnetic field is beyond the scope of this paper. Then, we refer the reader to the literature that deals with the dynamo processes, global and local, which are believed to be at the origin of the emergence, dynamics and cancellation of this magnetic field (e.g., [47–51]). Of interest for this work are the convective transport mechanisms of magnetic elements and the consequent magnetic pattern formation that is intimately linked to the boundaries of the underlying convective structures. In particular, we refer to the pattern formed by magnetic elements that are swept at the edges of convective structures, which leads to the formation of an irregular quasi-polygonal magnetic network that outlines the boundaries of convective cells [37–40].

Once we have the numerical tools to detect the PMVs, we study statistical properties that will be used to investigate the multiscale properties of the convective structures highlighted by the associated magnetic pattern. In this work we make use of the largest dataset of quiet-sun magnetograms used so far with the PMV algorithm to extend the statistics of previous works both in terms of spatial and temporal scales. PMVs were detected in the range 2–10 Mm in [39] and in the range 10–60 Mm in [40]. In this work we access the spatial scales from 1 to 80 Mm. Furthermore, we compare PMVs from SC23 and SC24 and we apply the technique to magnetograms from the Helioseismic and Magnetic Imager instrument for the first time.

## 2. Materials and Methods

### 2.1. Datasets

With the aim of identifying PMVs representative exclusively of photospheric convective patterns we selected quiet-sun line-of-sight magnetograms acquired during periods characterized by a low activity, typically associated with periods of solar minimum. This choice means that the possible effect of extended active regions on the magnetic pattern of convective origin is negligible. Data sets of quiet-sun magnetic images used in this work, and gathered from three different instruments on board three different space missions, i.e., SOHO, SDO and HINODE, are summarized in Table 1.

**Table 1.** For each instrument the table reports the number of analyzed magnetograms, the beginning and end of the period used to select the quiet-sun magnetograms, the pixel scale of the magnetograms, the size of the magnetograms in pixels and the total number of identified PMVs.

|  | SOHO/MDI | SDO/HMI | HINODE/SOT |
|---|---|---|---|
| Number of magnetograms | 191 | 500 | 1 |
| Start | 1996-08-01 | 2010-05-12 | 2007-03-10 |
| End | 1997-02-28 | 2019-05-27 | 2007-03-10 |
| Pixel scale (asec/pixel) | 0.620 | 0.504 | 0.150 |
| Size | $1024 \times 1024$ | $1024 \times 1024$ | $2047 \times 994$ |
| Detected PMVs | 133,569 | 183,359 | 942 |

In detail, the line-of-sight magnetograms were acquired from the following sources.

1.  The Solar Oscillations Investigation/Michelson Doppler Imager (SOI/MDI) instrument [52], on board the SOHO satellite, collected solar images from 1996 to 2011 using the spectral line Ni I 676.78 nm (e.g., [53,54]). The images were recorded by a $1024 \times 1024$ CCD camera in two spatial resolution modes: full disk and high-resolution of the central part of the disk. In this work, 191 original high-resolution images acquired during the solar activity minimum between SC 22 and SC 23 are used, covering a period of 7 months from 1 August 1996 to 28 February 1997. The images have a field of view of $11 \times 11$ arc min$^2$ with a plate scale of 0.625 arcsecond per pixel and a (diffraction-limited) resolution of 1.25 arc min (see http://soi.stanford.edu/sssc/progs/mdi/calib.htmlformoreinformationaboutdata (accessed on 1 July 2021)).

2. The Helioseismic and Magnetic Imager (HMI) is part of the NASA Solar Dynamics Observatory (SDO) mission [55]. HMI provides stabilized one-arcsecond-resolution full-disk Doppler velocity, line-of-sight magnetic flux, and continuum proxy images every 45 s, and vector magnetic field maps every 90 or 135 s depending on the image frame sequence selected. The solar image nearly fills the 4096 × 4096 pixel CCD camera allowing a pixel scale equal to 0.5 arcsecond per pixel [56,57]. HMI quiet magnetograms used in this work are the central sub-array of the original 4096 × 4096 full-disk images and they cover the whole mission from May 2010 to May 2019 (see http://jsoc.stanford.edu (accessed on 1 July 2021) for more information about data).

3. The Narrowband Filter Imager (NFI) of the Solar Optical Telescope (SOT) is on board the Hinode satellite [58]. The observation was recorded on 10 March 2007 between 11:37 and 14:34 UT, and views a portion of solar photosphere of 302 × 162 arcsecond$^2$ at the disc center. The original FITS header declares a different plate scale of 0.1476 arcsec/pixel and 0.1585 arcsec/pixel along x and y, respectively. Assuming for the solar radius the estimate of 696,342 ± 65 km [59] and the corresponding apparent radius of 959.6 ± 0.1 arcseconds [60], this difference corresponds to $\simeq$7.9 km, which has a negligible effect on the PMV identification algorithm and is therefore not taken into account in the analysis (see http://sdc.uio.no/sdc/ (accessed on 1 July 2021) for more information about data).

*2.2. PMV-Finding Algorithm*

The PMV-finding procedure we adopt in this work is described in full detail in [40] (henceforth BSD14). The procedure is based on that proposed by [61] (henceforth AM98). Compared to the algorithm proposed by AM98, our climbing algorithm has been significantly improved, making the BSD14 procedure faster. Indeed, this algorithm produces results faster then the conventional algorithm proposed by AM98, and actually we have reduced the time required for the PMV identification (about 25 times shorter than in AM98). The reduction of calculation times was useful in the testing phase allowing a greater number of analysis runs.

The steps of the PMV-finding procedure are as follows: 1. binarization of a magnetogram by means of the classification of pixels as magnetic or non-magnetic; 2. detecting empty region in the magnetic pattern; 3. recursive growth of the empty circles; 4. labeling of the circles for the definition of the single PMV.

To understand the PMV-finding procedure we start from the definition of distance field (*DF*) reported in AM98. This in turn is based on the empty sphere method [62]. Our scalar distance field $D : L^2 \rightarrow R$ is analogously defined, in a 2D space, as the distance of a given point $x$ to the nearest magnetic element M:

$$D(x) = min_j\{|x - M_j|\} \qquad (2)$$

where $M_j, j = 1, 2, \ldots, N$ are the sites of the magnetic elements highlighted by the binarized magnetogram. The climbing algorithm of AM98, based on the path along the monotonically increasing $\nabla DF(x, y)$ is replaced by a faster algorithm in which the climbing is carried out only on the centers of circles defined by a recursive algorithm of circle packing (Figure 1, Panel c). In each step a circle is drawn centered on the maximum of the current *DF* and having a radius equal to the local *DF* value (i.e., extending the radius up to the touch with the nearest magnetic structure). In the following step we insert the last generated circle as a fictitious magnetic area in the previously used binarized image and we calculate a new *DF*. Likewise in the Apollonian circle packing [63,64], a sequence of circles $C_1, C_2, C_3, \ldots$ having non-crescent diameters (therefore circles whose diameters are smaller than or at most equal to the diameters of the circles of the previous iteration) and filling the regions "empty" of magnetic elements is recursively generated. The iteration is stopped when the current $C_i$ radius $R$ is smaller than an assigned $R_{min}$ (please see BSD14 for details). The climbing algorithm is then executed only for the centers of the filling circles, saving a noteworthy amount of CPU time.

In Figure 1 an example of the complete process of the circle packing algorithm coverage relative to a typical SOHO/MDI magnetogram is shown.

The rectangular region (a) shows a sub-sample of an original magnetogram. To define the rectangular magnetic structures (region b) we used a fixed threshold T equal to 3 times the standard deviation ($\sigma$) of the absolute magnetic field values in the selected sub-region. In detail, the thresholds are 16 G, 15 G and 98 G for SOHO/MDI, SDO/HMI and HINODE/SOT, respectively. Please refer to BSD14 for an in-depth discussion on the projection effects of the magnetic field ($B_{LoS} = B_{normal}$) and on the choice of the threshold values. The rectangular region (c) is obtained from the application of the adaptive Apollonian circle packing procedure to cover the void area and the labeling of each circle center to the appropriate PMV by the climbing algorithm. Different colors identify different PMVs. Finally, the rectangular region (d) shows an estimation of PMV borders.

For the research of the PMVs, above reported, we used the same threshold for any analyzed magnetogram, i.e., T = $3\sigma$. We have seen that this value is valid for two of the datasets, i.e., SDO/HMI and SOHO/MDI, and the derived distributions are very similar (refer to the "Results and Discussion" section).

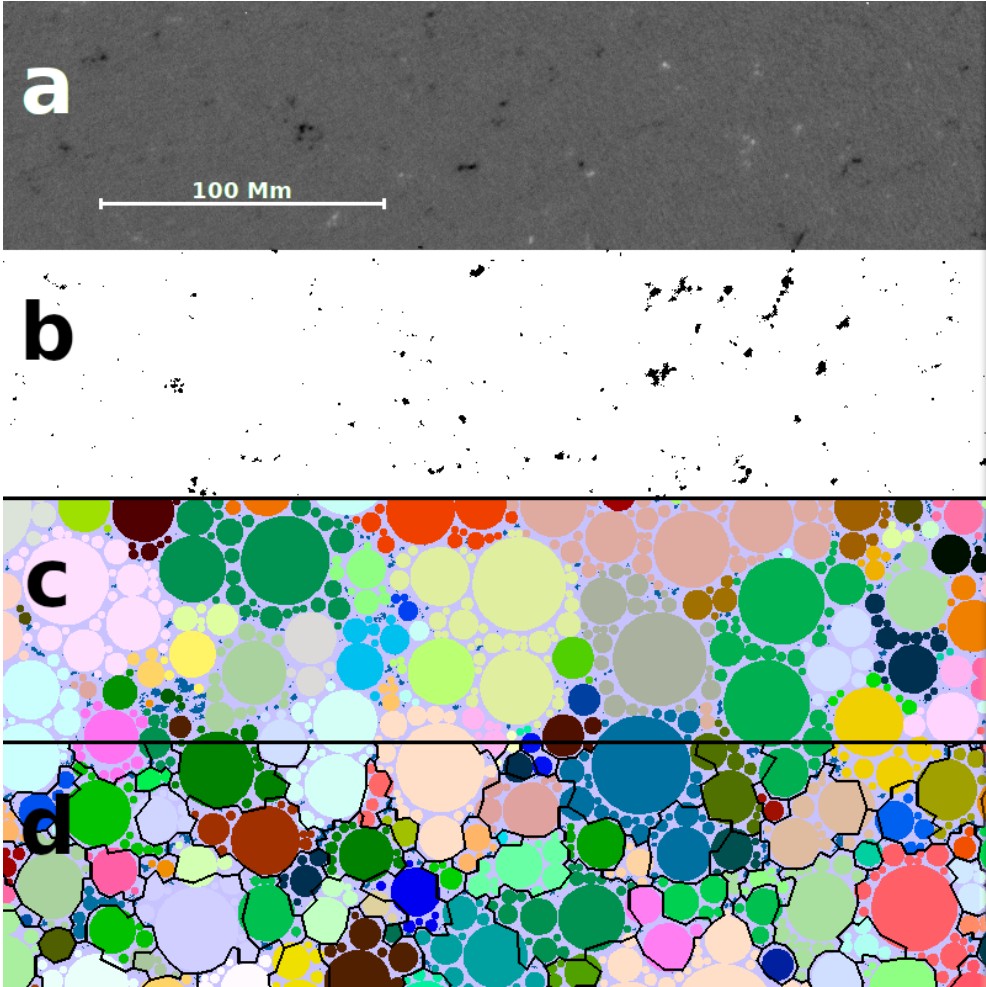

**Figure 1.** The circle packing algorithm on a SOHO/MDI magnetogram: (**a**) the original magnetogram, (**b**) the binarized version of the magnetogram, (**c**) the circle filling result (same color means association with the same PMV), (**d**) PMV borders (shown with solid black line).

In the case of Hinode SOT/SP we used a single high-resolution magnetogram that was sufficiently large enough to allow a statistically accurate analysis of the PMVs present in it. We refer to [39] (henceforth BGS13) for the details regarding the magnetogram used. Here we recall only the main characteristics of the magnetogram. We underline that, unlike

the void-finding algorithm used in BGS13, in this work we used for consistency the same fast procedure just described in this section that was applied to the other two datasets.

To study the effect of the threshold in the detection of PMVs in the high-resolution magnetogram, we used different thresholds. In fact, we expect that as the threshold increases, the regions emptied by magnetic elements tend to increasingly resemble the PMVs obtained from magnetograms with lower spatial resolution and magnetic sensitivity (i.e., SOHO/MDI and SDO/HMI). This is because as the threshold increases, the smaller PMVs highlighted by the lowest magnetic field values will disappear. To better explain what has just been discussed, we report in Figure 2 the result of the application to the Hinode SOT/SP magnetogram of our procedure using two different thresholds, $3\sigma$ and $6\sigma$, respectively. The effect discussed above is clearly seen. As the threshold value increases, the emptied regions outlined by the weaker fields tend to disappear and aggregate into larger regions. From a point of view of the distribution of PMVs as a function of size, this effect involves a gradual decrease in the measured slope and an approach to the slope observed for the larger scales, i.e., SDO/HMI and SOHO/MDI, due to the fact that the magnetograms of these two missions are less spatially resolved and less sensitive than the magnetogram derived from Hinode observations.

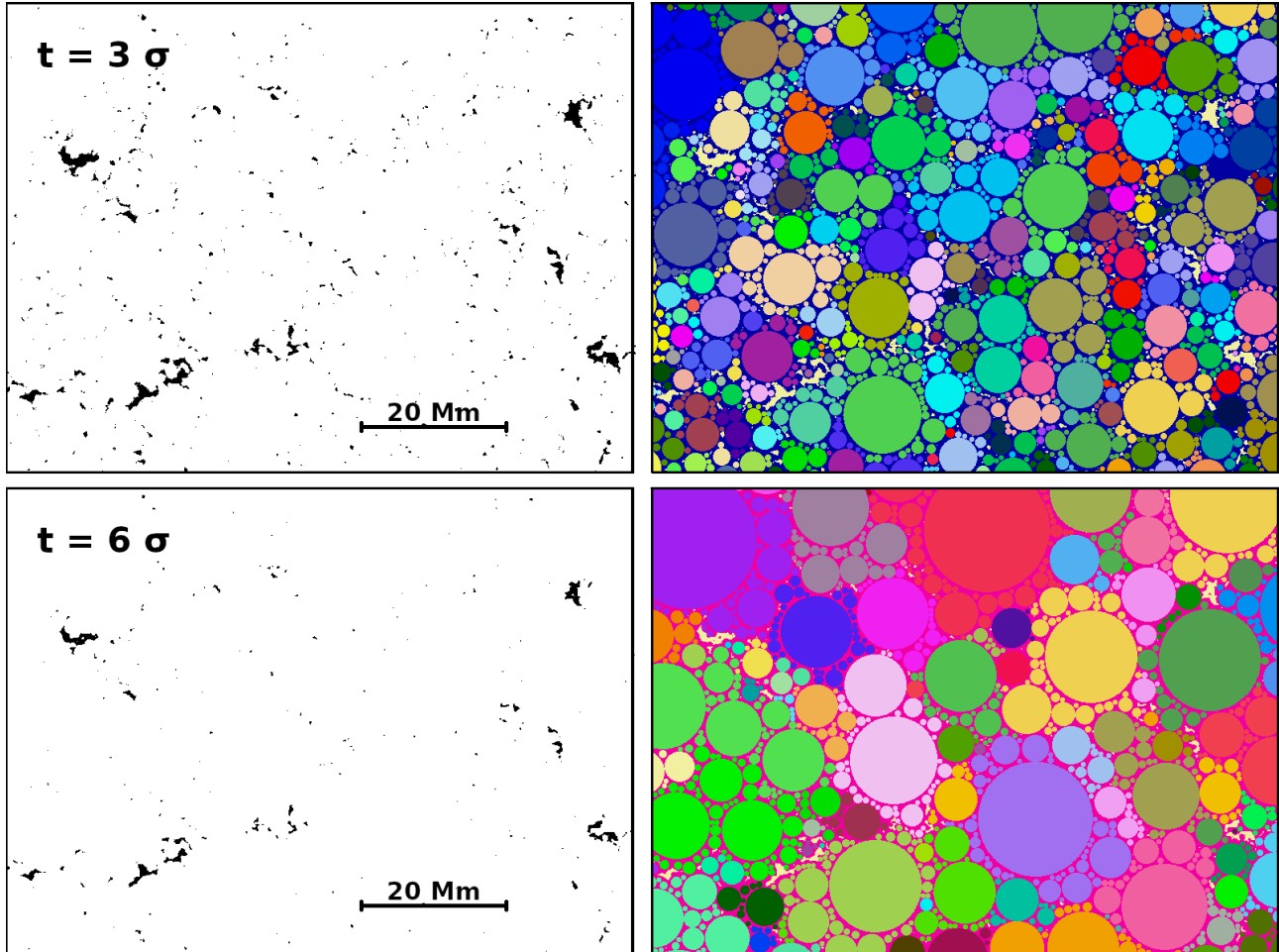

**Figure 2.** We show the result of the application to the Hinode SOT/SP magnetogram of our procedure using two different thresholds, $3\sigma$ (**upper panels**) and $6\sigma$ (**lower panels**). As the threshold value increases, the emptied regions outlined by the weaker fields tend to disappear and aggregate into larger regions.

## 3. Results and Discussion

The application of our PMV-finding procedure to 692 magnetograms found and labeled 317,870 PMVs. The equivalent diameter, i.e., the diameter of the circle of equal area of the detected PMV, was calculated and the PMV distribution is reported in Figure 3. The distribution of PMVs corresponds to what in BSD14 is called Void Size Distribution (or VSD). The present analysis confirms what is reported and discussed in BSD14. In particular, the PMV distribution is described by a near-exponential function in the observed range. Indeed, if we pass a best-fit line in the PMV counts derived from the Hinode data (for T = $6\sigma$) and in the averaged values of the counts derived from the SOHO/MDI and SDO/HMI magnetograms, we obtain for the line (black dash-dot line in Figure 3) the slope $S = -0.058 \pm 0.002$, which gives the exponential constant in the equation describing our PMV distribution:

$$PMV_{cd} \sim 10^{-0.058 \times E_D} \tag{3}$$

where $PMV_{cd}$ is the PMV density in $Counts/arcsec^2$ and $E_D$ is the PMV equivalent diameter in Mm. For the sake of completeness, we report that the residual distribution between the observed distribution and the exponential fit shows systematics that will be the subject of a future study. Indeed, as found in BSD14, the PMV distributions from SOHO/MDI and SDO/HMI data show a break at about 35 Mm. This scale is linked to one of the sources of systematics once a single exponential fit is performed on the entire range of spatial scales. The regime at large spatial scales stands up to 80 Mm without finding any preferential scales. Furthermore, in this work we find that the smaller spatial scales (Hinode data for T = $6\sigma$) seem to fit better with the trend for the larger spatial scales (>35 Mm). Nevertheless, as discussed in BSD14, the break at 35 Mm could be related to a convective instability scale able to inject downflow plumes. A possible instability mechanism that could support such a scale has been described in [65]. These plumes could become a new site for magnetic field concentrations, reducing the number of voids with equivalent diameters larger than 35 Mm. Further investigation is needed to support this hypothesis.

The statistic presented in this work is much greater than in BSD14. Indeed, present data cover two solar cycles, and the SOHO/MDI and SDO/HMI magnetograms refer to regions of quiet-sun relative to SC 23 and SC 24, respectively. Thus, this underlines how the dimensions of the large (i.e., equivalent diameter > 3 Mm) PMVs are very similar in the two cycles.

The analysis of the Hinode SOT/SP high-resolution magnetogram performed with the same PMV-finding procedure used to study large scale ranges allows us to extend the distribution up to the equivalent size of about 1 Mm. Apart from expected problems of connection between the two distributions, no particular organization scales are highlighted.

Furthermore, the near-exponential function that describes the VSD can also be used to describe other physical quantities in systems characterized by turbulent convection. For instance, the probability density of normalized temperature fluctuations is close to an exponential in the hard-turbulence regime ($Ra > 10^8$) [13], which is the case of photospheric turbulent convection ($Ra \approx 10^{11}$) [66], while it transits to a Gaussian in the soft turbulent convective regime ($Ra < 10^7$).

The absence of particular features in the distribution in the 1–80 Mm range rules out the presence of special scales (i.e., supergranulation or mesogranulation) when the magnetic pattern formed by surface convective flows is used to identify convective cell borders. This result supports the multiscale hypothesis of turbulent convective flows on the solar photosphere reported by several authors [39,40,67,68].

As clearly described in [67], theoretical models had anticipated the idea that convective structures, resulting from solar plasma flows, show a smooth spectrum, although the canonical literature typically refers to distinct scales of solar convection, such as granulation, mesogranulation and supergranulation. The multiscale view has been supported over time, not only in the papers just mentioned, but also in papers that studied the properties of convective velocity or intensity patterns at different scales [69,70], or introduced the

concept of 'trees of fragmenting granules' to study families of replicated structures at different scales starting from single granules [71].

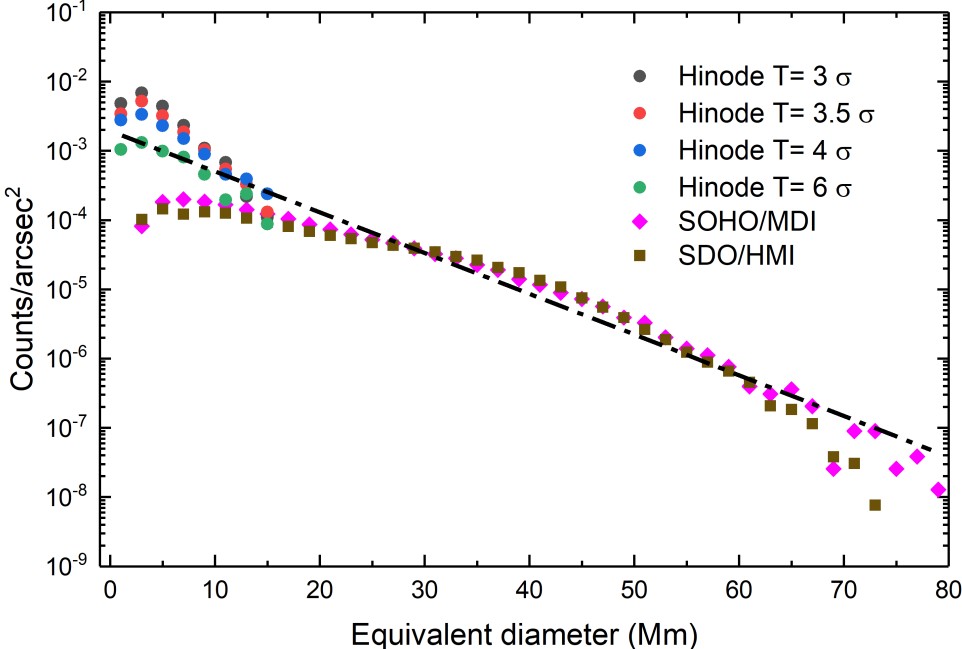

**Figure 3.** Distribution of PMV number versus equivalent diameter of voids. The plot shows the number density of PMVs obtained with our void detection algorithm. The different curves were obtained using different LoS magnetograms and magnetic thresholds. The circles denote the number density of "small"-scale PMVs detected in the Hinode magnetogram. The different colors refer to different magnetic thresholds. In detail, black circles refer to a magnetic threshold T = $3\sigma$, red circles refer to a magnetic threshold T = $3.5\sigma$, blue circles refer to a magnetic threshold T = $4\sigma$, and green circles refer to a magnetic threshold T = $6\sigma$. It is worth noting that as the threshold increases the number of small-sized HINODE PMVs decreases and the slope of the distributions approaches the slope derived for full-disk magnetograms. The pink diamonds and the brown squares denote the number density of "medium/large"-scale PMVs detected in SOHO/MDI (SC 22) and SDO/HMI (SC 23) magnetograms, respectively. The black dash-dot line, with slope $S = -0.058 \pm 0.0024$, represents the exponential behavior of PMV size distribution.

## 4. Conclusions

This work presents the most extensive statistics, both in the spatial scales studied and in the temporal duration, of photospheric regions substantially "emptied" of an intense magnetic field. By intense magnetic field we mean magnetic field values higher than a suitably defined threshold (see Section 2). Our method relies on the well-established hypothesis (e.g., [72]) that these "emptied" regions, which we have called PMVs or voids, are produced by magnetic elements that are swept to the boundaries of convective cells by superficial advection flows and underline their pattern. As described in Section 2.2, we identified the PMVs in the selected quiet LoS magnetograms using a PMV finder based on a fast climbing algorithm.

For PMV statistics (number density versus effective diameter) we found remarkable agreement with previous void analysis and with theoretical predictions that suggested the absence of preferential scales in the organization of convective structures at the surface of the Sun.

Another important aspect of our methodology is that this behavior is extended in the spatial range 1–80 Mm. This range covers all the scales classically referred to as "typical" of the photospheric convective organization, i.e., granulation, mesogranulation, supergranulation and giant cells. This multiscale behavior therefore seems to be typical

of the turbulent convection of our star, as anticipated on theoretical grounds from the very beginning [67] and consequently supports the multiscale hypothesis of turbulent convection in stars.

A further important aspect to underline is the difference between turbulent convection and classical turbulence (for example the turbulence developed in terms of a hierarchy of scales in which the plasma velocity fluctuations follow the Kolmogorov scaling characterized by a power spectrum with a slope of $-5/3$). In this regard, we underline that the present work is focused on the analysis of the dimensions of convective cells in a range of scales far from the typical scales in which a local turbulence model can be applied, i.e., a scale of the order of the pressure scale height, 150 km. We refer to the paper [73] for an in-depth discussion on the nature of solar granulation, and in general of stellar convection, in terms of turbulence models.

Conclusions on whether the distribution we found is compatible or not with the presence of particular asymptotic regimes that are supposed to occur at very high $Ra$ and which are connected to heat transfer efficiency of the turbulent convection are more difficult to draw. The arduous future task will be that of combining the uncertainties in linking $Nu$ on $Ra$, in physical systems characterized by stratified plasma in a magneto-convective regime and very high $Ra$, to estimate the global efficiency in convective energy transport by refining the sophisticated models currently available but which, unfortunately, have computational domains that probe a range of scales that are still too small compared to the solar case.

**Author Contributions:** Conceptualization, F.B. and S.S.; methodology, S.S., F.B and D.D.M.; software, S.S.; validation, F.B., D.D.M. and L.G.; formal analysis, all; investigation, all; resources, S.S. and F.B.; data curation, S.S.; writing—original draft preparation, F.B. and S.S.; writing—review and editing, all; supervision, F.B. and S.S.; funding acquisition, F.B. All authors have read and agreed to the published version of the manuscript.

**Funding:** This research was partially funded by the Italian MIUR-PRIN grant 2017APKP7T on Circumterrestrial Environment: Impact of Sun–Earth Interaction and by the European Union's Horizon 2020 research and innovation program under grant agreement No. 824135 (SOLARNET).

**Institutional Review Board Statement:** Not applicable.

**Informed Consent Statement:** Not applicable.

**Data Availability Statement:** Not applicable.

**Acknowledgments:** SOHO is a project of international cooperation between ESA and NASA. The authors thank the MDI team for MDI data. SDO/HMI data are courtesy of NASA/SDO and the HMI science teams. Hinode is a Japanese mission developed and launched by ISAS/JAXA, with NAOJ as a domestic partner and NASA and STFC (UK) as international partners.

**Conflicts of Interest:** The authors declare no conflicts of interest.

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
