# Peer review of "Stellar Turbulent Convection: The Multiscale Nature of the Solar Magnetic Signature"

_atmosphere, doi:10.3390/atmos12080938_

Round 1

Reviewer 1 Report

Generally, it looks good. A few modifications/additions are needed:

(1) Add more technical details to explain the following (rather than point the reader to a reference):

    line 126: "DF distance field"

    line 133 "non-crescent diameters"

(2) What is meant by 'exponential function' to describe Fig. 3? Is it exp(-bx) or exp(-bx2), or something else?

(3) Are there power laws suggested by the data shown in Fig. 3, i.e., what is α in (diameter) in different sections of the curve?

(4) In regard to turbulence, how does Fig. 3 relate to the Kolmogorov law ~k-5/3? Since the PMVs are caused by turbulent convection, some more commentary on what they tell us about the underlying turbulence would be useful.

(5) Be more careful with use of 'convection' (a noun) and 'convective' (an adjective). Also, 'convention' is used once, when 'convection' is meant. Similarly, care is needed in the use of 'turbulence' (a noun) and 'turbulent' (an adjective).

Author Response

We thank you for the careful reading of the manuscript. Below are the answers to your comments (bold).  Corrections in the amended version of the latex manuscript file are in bold factor.

Abstract. Line 5. Acronym of ‘CZ’ should be defined, i.e. CZ à CZ (convective zone)

Done.

Page 4, line 135, “The climbing algorithm is then executed only … saving a noteworthy amount of CPU time.” Please describe the comparison in detail, using numbers [cpu-h], for typical cases.

We have added comments on this at the beginning of section 2.2

Page 6, line 170. It seems there is a strike-out line which may be a trace of editing.

Corrected

Page 7, line 191. There is a typo, “Ra~10^{1}1”

Corrected

Reviewer 2 Report

The manuscript reports the statistical analysis application, especially multi-scaling analysis of convections in stars. The PMVs (Photospheric magnetic voids) method which is performed by the authors shows a good agreement with conventional void analysis. The manuscript shows the procedure in clear manner, therefore the manuscript should contribute the community of atmosphere research. For this, I would recommend the manuscript to be published in the journal, Atmosphere, after some minor comments and questions.

Abstract. Line 5. Acronym of ‘CZ’ should be defined, i.e. CZ  CZ (convective zone)

Page 4, line 135, “The climbing algorithm is then executed only … saving a noteworthy amount of CPU time.” Please describe the comparison in detail, using numbers [cpu-h], for typical cases.

Page 6, line 170. It seems there is a strike-out line which may be a trace of editing.

Page 7, line 191. There is a typo, “Ra~10^{1}1”

Author Response

We thank you for the careful reading of the manuscript. Below are the answers to your comments (bold).  Corrections in the amended version of the latex manuscript file are in bold factor.

Generally, it looks good. A few modifications/additions are needed:
Add more technical details to explain the following (rather than point the reader to a reference):
    line 126: "DF distance field"
    line 133 "non-crescent diameters"

We have added explanations and technical details on the points raised by referee in section 2.2

What is meant by 'exponential function' to describe Fig. 3? Is it exp(-bx) or exp(-bx2), or something else?

We presented and discussed the exponential fit in section 3. We also added it in Fig. 3 in order to clarify the meaning of exponential function.

Are there power laws suggested by the data shown in Fig. 3, i.e., what is α in (diameter) in different sections of the curve?

The point raised is very interesting and has been discussed in a previous paper (referred to in the paper as BSD14). Indeed, if reported in log-log scale, the distribution can appear as describable by a double power law (albeit over a not particularly extensive extension of spatial scales). It is currently unclear what could introduce this effect. As discussed in this paper, again in section 3, the break at 35 Mm could be related to a convective instability scale able to inject downflow plumes. But this is a hypothesis that would require a different analysis and the use of models on quasi-global domains and ranges of convection scales not currently available.

In regard to turbulence, how does Fig. 3 relate to the Kolmogorov law ~k-5/3? Since the PMVs are caused by turbulent convection, some more commentary on what they tell us about the underlying turbulence would be useful.

We commented this point in section 4.

Be more careful with use of 'convection' (a noun) and 'convective' (an adjective). Also, 'convention' is used once, when 'convection' is meant. Similarly, care is needed in the use of 'turbulence' (a noun) and 'turbulent' (an adjective).

We reread the manuscript and corrected some points where confusion could arise between "name" and "adjective" in connection with the concepts of convection and turbulence.